# Bacteriostatic Activity and Resistance Mechanism of *Artemisia annua* Extract Against *Ralstonia solanacearum* in Pepper

**DOI:** 10.3390/plants14050651

**Published:** 2025-02-20

**Authors:** Tiantian Du, Weiping Zhu, Chenning Zhang, Xiaomin Liang, Yinghua Shu, Jingyun Zhou, Mengyu Zhang, Yuxuan He, Jincai Tu, Yuanjiao Feng

**Affiliations:** 1Key Laboratory of Agro-Environment in the Tropics, Ministry of Agriculture and Rural Affairs, South China Agricultural University, Guangzhou 510642, China; du15546436576@163.com (T.D.); zhangcn@stu.scau.edu.cn (C.Z.); xmliang@stu.scau.edu.cn (X.L.); shuyinghua@scau.edu.cn (Y.S.); zhangmy20220789@163.com (M.Z.); yuxuanhe0427@stu.scau.edu.cn (Y.H.); tujincai0211@163.com (J.T.); 2Guangdong Engineering Research Center for Modern Eco-Agriculture and Circular Agriculture, Guangzhou 510642, China; 3Department of Ecology, College of Natural Resources and Environment, South China Agricultural University, Guangzhou 510642, China; 4Institute of Chinese Agricultural History Heritage, South China Agricultural University, Guangzhou 510642, China; zwp312@126.com; 5Instrumental Analysis & Research Center, South China Agricultural University, Guangzhou 510642, China; zjy1341@126.com

**Keywords:** bacterial wilt, bio-control, antibacterial activity, enzyme activity, plant extracts

## Abstract

The destructive bacterial wilt disease caused by *Ralstonia solanacearum* leads to substantial losses in pepper production worldwide. Plant-derived pesticides exhibit advantages of high efficiency and broad spectrum when compared to traditional chemical pesticides. *Artemisia annua* and ‘Tai Jiao’ No. 1 were used as the experimental materials, and treated with 0.75 g·mL^−1^, 1.5 g·mL^−1^, and 3 g·mL^−1^ of *A. annua* extract and inoculated with *R. solanacearum* at a concentration of OD600 = 0.1 for 14 days. The inhibitory activity of *A. annua* extracts against *R. solanacearum,* as well as the disease index, defense enzyme activities, and defense-related substances contents of pepper seedlings were determined. The results showed that the Minimum Inhibitory Concentration (MIC) of the *A. annua* extract was 3 g·mL^−1^. As the concentration of *A. annua* extract increased, the extent of *R. solanacearum* cell crumpling intensified, accompanied by a gradual decline in its biofilm-forming ability. On the 14th day after treatment, the disease severity index and incidence rate were significantly reduced when the *A. annua* extract was applied at concentrations of 0.75 g·mL^−1^ and 3 g·mL^−1^. At both the 7th and 14th days after treatment, the application of *A. annua* extract at concentrations of 0.75 g·mL^−1^ and 3 g·mL^−1^ led to enhanced activities of superoxide dismutase (SOD), peroxidase (POD), and catalase (CAT) in peppers at different stages. Simultaneously, it reduced the levels of malondialdehyde (MDA) and hydrogen peroxide (H_2_O_2_), effectively scavenging reactive oxygen species and alleviating cellular lipid peroxidation. Furthermore, the extract increased the activities of polyphenol oxidase (PPO) and phenylalanine ammonia-lyase (PAL), as well as elevated the contents of soluble protein, flavonoids, and total phenols, ultimately enhancing the disease resistance of peppers. Considering the development costs, the application of *A. annua* extract at a concentration of 0.75 g·mL^−1^ demonstrates great potential for green control measures against bacterial wilt in peppers.

## 1. Introduction

Pepper serves as a prominent cash crop in China, with a cultivation area exceeding 1,533,300 hm^2^ in 2021. This figure accounts for 40% of the global cultivation area, thereby establishing it as the most extensively grown vegetable in China [1,2]. Pepper bacterial wilt, caused by *Ralstonia solanacearum*, is a soil-borne bacterial disease distinguished by its broad host adaptability, potent latent infectivity, and significant genetic variability, which pose severe impediments to the advancement of the pepper industry [3,4,5]. *Ralstonia solanacearum* can enter the plant through wounds on the roots and stems of peppers. Upon reaching the pepper roots, the bacterium initially attaches to the cortical cells. After proliferating extensively in the roots, it continuously invades the xylem and vascular bundles of the host plant and secretes a large amount of exopolysaccharides (EPS) in the vascular bundles. This leads to wilting of peppers and even the death of the entire plant [6].

Worldwide, the prevention and control of bacterial wilt in peppers rely on comprehensive measures, such as disease-resistant breeding, optimization of cultivation conditions, improvement of fertilization methods, use of chemical agents, and crop rotation [7,8]. However, the prevention and control effects are limited, and there is an urgent need to explore other effective measures for controlling bacterial wilt in peppers [9,10,11]. Plant-derived pesticides exhibit advantages of high efficiency and broad spectrum when compared to traditional chemical pesticides [12]. They have played a significant and positive role in reducing chemical pesticide use, ensuring agricultural product safety, maintaining ecological balance, and implementing green pest control for specialty crops. As of November 2021, in China, there were a total of 727 registered pesticide varieties, among which 30 were plant-derived pesticides, accounting for 4% of the total [13]. However, due to the limited number of plant-derived pesticides and the lack of those specifically developed to control bacterial wilt in peppers, this field still requires further development.

The dried aerial parts of the plant Artemisia annua contain artemisinin, its main compound, which has strong antimalarial properties and is a key ingredient in global antimalarial drugs [14,15,16]. The main chemical components of *A. annua* include sesquiterpenes, flavonoids, coumarins, volatile oils, and polysaccharides [17]. At a concentration of 0.1 g·mL^−1^, the aqueous extract of *A. annua* shows a 100% inhibition rate against *Phytophthora capsici* and *Mycogone pemiciosa* [18]. Similarly, the ethanol extract of *A. annua* has bacteriostatic activity against the pathogen of tomato late blight, and its MIC is 15.63 mg·mL^−1^ [19]. At a concentration of 60.0 μg·mL^−1^, the extract of *A. annua* shows inhibitory effects on the single-spore isolate of *Peronophythora litchi* [20]. Research shows that *A. annua* possesses strong bacteriostatic activity and has potential for the development of plant-based pesticides.

*R. solanacearum* possesses the ability to survive for extended periods in soil or plant residues. The combination of its robust infectivity and broad host range poses significant challenges in achieving complete control of bacterial wilt in agricultural settings. Moreover, as of the present date, no plant-derived fungicide has been developed specifically for the management of pepper bacterial wilt [21,22,23]. Chinese medicinal herbs are characterized by their extensive variety, ease of availability, diverse chemical compositions, and immense potential for research and development [24,25]. *A. annua* extracts display good antibacterial and antiviral activities against *Escherichia coli*, *Staphylococcus aureus*, and viruses in the *Coronaviridae*, *Filoviridae*, and *Rhabdoviridae* families [26,27,28,29]. Therefore, the development of plant-based fungicides specifically aimed at preventing and controlling bacterial wilt in peppers can provide new options for the future management of this disease. Given the remarkable antimicrobial capabilities of *A. annua*, we formulate the hypothesis that it may exhibit inhibitory effects against *R. solanacearum* as well. This study intends to clarify the bacteriostatic activity of *A. annua* extract against *R. solanacearum* and its influence on the disease-resistance metabolism in peppers. For this purpose, we measured the in vitro bacteriostatic effect of *A. annua* extract on *R. solanacearum* and its impact on the changes in defense enzyme activity within peppers. The objectives of this study’s findings are to furnish theoretical backing for the development of *A. annua* as a botanical fungicide for the prevention and management of bacterial wilt in peppers.

## 2. Results

### 2.1. The Inhibitory Effect of A. annua Extract on R. solanacearum Activity

The extract of *A. annua* exhibited inhibitory effects on *R. solanacearum*, with an EC_50_ value of 2.66 g·mL^−1^ (Table 1, Figure 1).

### 2.2. MIC Value

At concentrations of 0.19–1.5 g·mL^−1^, it is red, and at concentrations above 3 g·mL^−1^, the solution has no color change. As shown in Table 2, the minimum inhibitory concentration (MIC) value of *A. annua* extract is 3 g·mL^−1^.

### 2.3. Inhibition Curve of A. annua Extract

Figure 2 demonstrates the inhibitory impact of the *A. annua* extract on *R. solanacearum.* At a concentration of 3 g·mL^−1^, the extract slowed the growth of *R. solanacearum* within the first 24 h and subsequently halted its growth completely after 24 h, as compared to the blank control.

### 2.4. The Effect of A. annua Extract on the Biofilm of R. solanacearum

The formation of *R. solanacearum* biofilm was inhibited by the *A. annua* extract. In addition to a weak upward trend in biofilm formation for the 1.5 g·mL^−1^ treatment, there was still an inhibitory effect compared to the control. Overall, increasing the concentration of *A. annua* extract decreased the biofilm synthesizing ability of *R. solanacearum* (Figure 3).

### 2.5. The Effect of A. annua Extract on the Morphology of R. solanacearum

The morphological alterations of *R. solanacearum* were observed following treatment with *A. annua* extract at varying concentrations. In the control group (0 g·mL^−1^), the bacterial cells maintained their normal, turgid morphology without any observable deformations. However, treatment with *A. annua* extracts induced notable morphological alterations, including cell shrinkage, deformation, and surface depression. These morphological changes exhibited a concentration-dependent response, with more pronounced shrinkage, deformation, and depression observed at higher extract concentrations (Figure 4).

### 2.6. The Efficacy of A. annua Extract in Controlling Pepper Wilt Disease Caused by R. solanacearum

The extract from *A. annua* could significantly reduce the disease index and incidence of pepper bacterial wilt, which was attributed to *R. solanacearum*. On the 7th day, the disease index and incidence rate in the treatment solely involving *R. solanacearum* were 44.33 and 40.67%, respectively. In contrast, the treatments with 0.75 g·mL^−1^, 1.50 g·mL^−1^, and 3.00 g·mL^−1^ extracts of *A. annua* exhibited disease indices and incidence rates of 14.29 and 10%, 14.29 and 10%, and 0%, respectively. On the 14th day, the disease index and incidence rate for the treatment solely involving *R. solanacearum* were 66.67% and 78%, respectively. In comparison, the treatments with 0.75 g·mL^−1^, 1.50 g·mL^−1^, and 3.00 g·mL^−1^ extracts of *A. annua* exhibited disease indices of 25%, 42.85%, and 21.42%, respectively, and incidence rates of 30.33%, 44.33%, and 22%. These findings demonstrate that the application of *A. annua* extracts resulted in a notable decrease in both the disease index and incidence rate compared to the *R. solanacearum*-only treatment (Figure 5).

### 2.7. The Impact of A. annua Extract on the Content of Primary Metabolites in Pepper

*A. annua* extract had no significant impact on the soluble sugar content in pepper leaves (Figure 6a).

The application of *A. annua* extract had the capacity to elevate the soluble protein content within pepper leaves. On the 7th day, a comparison was made between pepper leaves treated solely with *R. solanacearum* and those treated with 0.75 g·mL^−1^ and 3 g·mL^−1^ of *A. annua* extract. The results showed that the soluble protein content in the leaves treated with the extracts increased by 20.98% and 19.20%, respectively, as illustrated in Figure 6b.

The free amino acid content of the R treatment was significantly higher than that of the other four treatments at day 7, but the difference disappeared at day 14. *A. annua* extract has little effect on the free amino acid content in pepper leaves (Figure 6c).

### 2.8. The Impact of A. annua Extract on the Content of Defense System Substances in Pepper

#### 2.8.1. The Effect of *A. annua* Extract on the MDA Content in Pepper

*A. annua* extract can reduce the MDA content in pepper leaves. On the 7th day, the MDA content in pepper leaves treated with 0.75 g·mL^−1^, 1.5 g·mL^−1^, and 3 g·mL^−1^ of *A. annua* extract was observed to decrease by 3.96%, 1.29%, and 6.49% respectively, in comparison to leaves treated solely with *R. solanacearum*. Furthermore, on the 14th day, a significant effect was noted for the 0.75 g·mL^−1^ *A. annua* extract treatment, with a 22.49% reduction in MDA content compared to the *R. solanacearum*-only treatment (Figure 7).

#### 2.8.2. The Effect of *A. annua* Extract on the H_2_O_2_ Content in Pepper

*A. annua* extract can reduce the H_2_O_2_ content in pepper leaves, but the reduction effect diminishes over time. On the 7th day, the H_2_O_2_ content in pepper leaves was analyzed following treatments with 0.75 g·mL^−1^, 1.5 g·mL^−1^, and 3 g·mL^−1^ of *A. annua* extract, in comparison to leaves treated solely with *R. solanacearum*. The results indicated a decrease of 21.31%, 34.51%, and 9.97%, respectively, in the treated leaves (Figure 8).

#### 2.8.3. The Effect of *A. annua* Extract on the Total Phenol Content in Pepper

The total phenol content in pepper leaves was found to increase following treatment with *A. annua* extract. Specifically, on the 7th day, leaves treated with 3 g·mL^−1^ of *A. annua* extract exhibited a 12.56% increase in total phenol content compared to those treated solely with *R. solanacearum*. Additionally, on the 14th day, leaves treated with 1.5 g·mL^−1^ and 3 g·mL^−1^ of the extract showed respective increases of 5.53% and 7.71% in total phenol content (Figure 9).

#### 2.8.4. The Effect of *A. annua* Extract on the Flavonoid Content in Pepper

The flavonoid content in pepper leaves was observed to increase following treatment with *A. annua* extract, albeit without a clear dose-response relationship among various concentrations. Specifically, on the 7th day, leaves treated with 0.75 g·mL^−1^, 1.5 g·mL^−1^ (corrected from an apparent typo of 0.5 g·mL^−1^), and 3 g·mL^−1^ of *A. annua* extract exhibited increases in flavonoid content by 6.11%, 2.41%, and 1.85%, respectively, compared to those treated solely with *R. solanacearum* (Figure 10).

### 2.9. The Effect of A. annua Extract on the Activity of Enzymes Related to the Pepper Defense System

The SOD activity in pepper leaves was found to be enhanced by the application of *A. annua* extract. Specifically, on the 14th day, leaves treated with 0.75 g·mL^−1^, 1.5 g·mL^−1^, and 3 g·mL^−1^ of the extract exhibited increases in SOD activity by 157.61%, 90.04%, and 178.23%, respectively, compared to those solely treated with *R. solanacearum* (Figure 11a).

The POD activity in pepper leaves was observed to increase following treatment with *A. annua* extract. Specifically, on the 7th day, leaves treated with 0.75 g·mL^−1^, 1.5 g·mL^−1^, and 3 g·mL^−1^ of the extract showed POD activity enhancements of 5.97%, 221.17%, and 283.14%, respectively, in comparison to those solely treated with *R. solanacearum* (Figure 11b).

The application of *A. annua* extract was found to elevate the CAT activity in pepper leaves. Specifically, on the 7th day, leaves treated with 0.75 g·mL^−1^ and 1.5 g·mL^−1^ of the extract showed increases in CAT activity by 73.32% and 194.06%, respectively, compared to those solely treated with *R. solanacearum*. Furthermore, on the 14th day, a 58.57% increase in CAT activity was observed in leaves treated with 3 g·mL^−1^ of the extract, in comparison to the control group treated solely with *R. solanacearum* (Figure 11c).

At various time points, different concentrations of *A. annua* extract have been observed to enhance the PPO activity in pepper leaves. On the 14th day in particular, leaves treated with 0.75 g·mL^−1^, 1.5 g·mL^−1^, and 3 g·mL^−1^ of the extract exhibited increases in PPO activity by 141.86%, 35.98%, and 211.87%, respectively, compared to those solely treated with *R. solanacearum* (Figure 11d).

The application of *A. annua* extract has been found to elevate the PAL activity in pepper leaves, with the enhancement effect gradually intensifying over time. Specifically, on both the 7th and 14th days, leaves treated with 0.75 g·mL^−1^, 1.5 g·mL^−1^, and 3 g·mL^−1^ of the extract showed increased PAL activity to different extents when compared to those solely treated with *R. solanacearum* (Figure 11e).

## 3. Discussion

This study provides experimental and scientific evidence that the extract of *A. annua* can effectively inhibit *R. solanacearum*, with a MIC of 3 g·mL^−1^ (Table 2).

Protein virulence factors, such as extracellular proteases, extracellular polysaccharides, and biofilms, play pivotal roles in the pathogenicity of *Ralstonia solanacearum* [30,31,32,33,34]. Within a specific concentration range, both harmine and daphnretin have been shown to inhibit the formation of *R. solanacearum* biofilms [35,36]. In this study, the addition of *A. annua* extract at a concentration of 3 g·mL^−1^ and incubation for 24 h significantly suppressed the biofilm-forming ability of *R. solanacearum* (Figure 3). These results indicate that the inhibitory effect of *A. annua* extract on biofilm formation directly impacts the pathogenicity of *R. solanacearum*, consistent with findings reported in previous studies. Maintaining intact cellular morphology is essential for bacteria to perform normal physiological and biochemical processes. Under scanning electron microscopic examination, *R. solanacearum* displayed depression, denaturation, and shrinkage at a concentration of 1.5 g·mL^−1^, with these effects becoming more pronounced at a concentration of 3 g·mL^−1^ (Figure 4). Similar morphological abnormalities, including cell depression, denaturation, and shrinkage, were observed in *R. solanacearum* cells treated with volatile organic compounds (VOCs) and *Bacillus velezensis* [37,38].

The natural existence of plants inevitably leads to their encounters with a diverse array of pathogenic microorganisms in their environment. Through the protracted process of interacting and adapting to these pathogens, plants have progressively evolved intricate and highly sophisticated defense systems aimed at minimizing and mitigating the threats posed by these pathogens [39]. In the context of plant defense systems, a close correlation has been observed between primary and secondary metabolites, wherein the latter are utilized for defensive purposes [40]. Certain primary metabolites play crucial roles in these systems by serving as precursors for the biosynthesis of defensive secondary metabolites, regulating their production, or directly contributing to defense mechanisms [41]. The application of *A. annua* extract has been shown to increase the soluble protein content in pepper leaves (Figure 6b). The application of melatonin prior to tomato harvest enhances the resistance of the fruits to *Botrytis* cinerea through an increase in soluble protein content [42]. In the case of artificial inoculation with the pathogen responsible for brown spot disease, disease-resistant *Juglans regia* (Persian walnut) varieties exhibit a significant increase in soluble protein content compared to susceptible varieties [43].

Reactive oxygen species (ROS), including superoxide anion (·O^2−^), hydrogen peroxide (H_2_O_2_), hydroxyl radical (·OH), and others, can accumulate excessively in plants, leading to cellular damage and disruption of oxidative metabolism. The ROS scavenging system comprises enzymes such as SOD, POD, and CAT [44,45,46,47,48]. This study reveals that seven days after the application of *A. annua* extract, pepper plants showed a significantly lower incidence rate and disease index of bacterial wilt compared to plants treated only with the pathogen. However, by the 14th day, both the incidence rate and disease index exhibited slight increases (Figure 5). Our findings indicate that, in pepper plants treated with *A. annua* extract, the activities of SOD, POD, and CAT are elevated (Figure 11a–c), while the levels of MDA and H₂O₂ are decreased (Figure 7 and Figure 8). These changes are closely associated with decreased ROS accumulation and reduced membrane lipid peroxidation in pepper leaves. Similarly, the application of cis-abietinol has been shown to enhance the activities of CAT, PPO, PAL, POD, and SOD in tobacco roots at various stages post-treatment [49], aligning with the findings of this study.

The application of *A. annua* extract leads to a notable increase in PPO and PAL activities in pepper leaves. Among them, PAL shows the most pronounced and sustained elevation (Figure 11d,e). Additionally, this treatment enhances the total phenol and flavonoid contents in pepper leaves (Figure 9 and Figure 10). PPO serves as a pivotal enzyme in plants’ responses to both biotic and abiotic stressors, playing a fundamental role in plant defense mechanisms, modulating plastid oxidation levels, and participating in the phenylpropanoid pathway [50]. PAL functions as a central and rate-limiting enzyme in the biosynthetic pathways of phytoalexins, lignin, and phenolic compounds, thereby playing a crucial role in the phenylpropanoid metabolism of plants and being intimately associated with the generation of phenolic compounds [51]. Research shows that pathogen induction significantly upregulates PAL activity in plants, a response intimately tied to the mechanisms underlying systemic acquired resistance. This upregulation promotes downstream metabolic pathways, producing branch products such as total phenols and flavonoids that enhance plant resistance to pathogen-induced damage [52,53]. The application of *A. annua* extract can activate the immune defense mechanisms in peppers, effectively mitigating the progression of bacterial wilt disease and strengthening disease resistance. Similarly, studies have found that hesperidin application significantly increases PAL and POD activities in tobacco leaves while enhancing the levels of phenolics and flavonoids in tobacco’s secondary metabolites [54]. The application of iturin has been shown to boost the activities of PAL, POD, SOD, and CAT in strawberries, offering protection against gray mold disease [55]. Our findings are consistent with these results. During the assessment of defense enzyme activity, dynamic variations in their activities across different stages were observed. Regulation of these enzymes’ activities is governed by a multitude of genes, while post-transcriptional regulation is further influenced by a variety of factors, leading to notable discrepancies between the alterations in enzyme activity and the corresponding levels of gene expression [56]. In order to gain a deeper understanding of the mechanism by which the extract of *A. annua* augments disease resistance in peppers, future studies necessitate the conduct of transcriptome analysis on the sampled pepper leaves, with a focus on comparing the upregulation status of genes associated with defense enzymes.

The variations in antibacterial activity observed among plant extracts can be ascribed to differences in the concentrations of active components within these extracts [57]. In this experiment, a crude extract of *A. annua* was used, containing a complex array of chemical constituents, including sesquiterpenes, coumarins, alkaloids, and lignans [58]. This study represents a preliminary exploration of the inhibitory effects of *A. annua* extract on *R. solanacearum* and its role in enhancing pepper resistance. However, the specific monomeric chemical constituents responsible for its antibacterial activity have yet to be identified and warrant further investigation in future research.

## 4. Materials and Methods

### 4.1. Materials

The strain of *R. solanacearum* was provided by the Population Microbiology Research Center of the Microbiology Laboratory at South China Agricultural University and was isolated from *R. solanacearum* EP1 in Guangdong Province. EP1 was cultured in CPG solid medium and CPG liquid medium. For virulence testing of *R. solanacearum*, TTC medium was used, and the media were prepared according to the method described by Zhang Qian [59]. The pepper variety used was ‘Tai Jiao’ No. 1. *A. annua* was purchased from Guangzhou Haowanjia Pharmacy.

### 4.2. Preparation of the A. annua Extract

*A. annua* was extracted via an ultrasonic extraction method. Firstly, 320 g of *A. annua* was weighed and mixed with pure water at a ratio of 1 g of *A. annua* to 5 mL of pure water. Subsequently, the mixture was stirred for 1.5 h each time, filtered under vacuum suction, and this procedure was repeated three times. After that, the combined filtrates were dried with a rotary evaporator at 60 °C, dissolved in 200 mL of pure water, and stored in a refrigerator at 4 °C for subsequent use.

Take the preserved *A. annua* extract and prepare solutions with concentrations of 0.75, 1.5, and 3 g·mL^−1^ using tap water for pot experiments.

### 4.3. Determination of the Bacteriostatic Activity of A. annua Extract

#### 4.3.1. Solid Plate Assay of *A. annua* Extract for Inhibiting *R. solanacearum*

We prepared aqueous extracts of *A. annua* in sterile water to create solutions with concentrations ranging from 2 g·mL^−1^ to 16 g·mL^−1^ (specifically, 2 g·mL^−1^, 6 g·mL^−1^, 10 g·mL^−1^, 12 g·mL^−1^, and 16 g·mL^−1^). These solutions were then sterile-filtered using a filter membrane. In a laminar flow hood under sterile conditions, CPG solid medium was allowed to cool to 50 °C before being mixed with *R. solanacearum* to achieve a final OD600 of 0.05. The mixture was then poured into petri dishes. After the medium had fully solidified, holes of 0.5 cm in diameter were punched into the CPG medium. Following this, 40 μL of aqueous extract of *A. annua* at varying concentrations and 40 μL of sterile water (with each treatment replicated three times) were sequentially dispensed into the respective holes. Following a 2-h pre-diffusion period at 4 °C, the cultures were incubated at 28 °C under controlled environmental conditions until full colonization of the medium was achieved. The inhibition zone size was observed, and its diameter was measured using the cross-hair method to calculate the average. The diameter of the inhibition zone serves as an indicator of antibacterial activity, categorized into four grades: absence of inhibition zone denoted as “-”, diameter less than 1.0 cm as “+”, diameter ranging from 1.1 to 1.5 cm as “++”, and diameter exceeding 1.5 cm as “+++”. EC_50_ value calculation was performed using GraphPad Prism 8.

The size of the inhibition zone = The diameter of the inhibition zone − 0.5 cm

#### 4.3.2. The Determination of Minimum Inhibitory Concentration (MIC)

With slight modifications, referring to the method of Li et al. [60]. A serial double-dilution method was employed to prepare six concentration gradients of the aqueous extract of A. annua in TTC liquid medium, namely 0.19 g·mL^−1^, 0.38 g·mL^−1^, 0.75 g·mL^−1^, 1.5 g·mL^−1^, 3 g·mL^−1^, and 6 g·mL^−1^. Then, 150 μL of the TTC liquid medium with different concentrations of the *A. annua* extract was dispensed into a 96-well microtiter plate. Next, 50 μL of *R. solanacearum* suspension with an OD600 value of 0.05 was added to each well. TTC medium was used as the negative control. After sealing, the plates were incubated on a shaker at 28 °C and 100 rpm for 24 h, and each treatment had three replicates. Monitor the growth of *R. solanacearum*, and determine the MIC as the point where no red coloration is observed.

#### 4.3.3. Measurement of Inhibition of *R. solanacearum* Growth

To prepare liquid CPG media with final concentrations of 0, 0.19, 0.38, 0.75, 1.5, and 3 g·mL^−1^, the aqueous extract of *A. annua* was added. Then, the resulting solution was sonicated to ensure uniform dispersion. The *R. solanacearum* bacterial suspension was cultured until it reached an OD600 value of 0.05. Subsequently, 200 μL aliquots of liquid media with different concentrations were dispensed into a 96-well plate. The experiment included a negative control (the bacterial suspension alone) and a blank control (CPG liquid medium). The cultures were incubated at 28 °C and 100 rpm, and their absorbance at a wavelength of 600 nm was measured at 0, 4, 8, 12, 24, 36, and 48 h. Each treatment was carried out in triplicate, and the mean values were calculated for constructing the antibacterial activity curve.

#### 4.3.4. Measurement of Biofilm Formation by *R. solanacearum*

To prepare liquid media with final concentrations of 0, 0.75, 1.5, 3, and 9 g·mL^−1^, add the extract of *A. annua* to liquid CPG medium. Culture the bacterial suspension of *R. solanacearum* until the OD600 reaches 0.05. Then, add the bacterial suspension to the media of different concentrations and mix uniformly. Aliquot 200 μL of the mixed solution into each well of sterile 96-well culture plates, with four replicates for each concentration. Incubate the plates at 28 °C without shaking for 24 h and 48 h, respectively. After incubation, carefully aspirate the cultures. Then, assess biofilm formation by *R. solanacearum* using the crystal violet staining method as described by Zhou et al. [61].

#### 4.3.5. Morphological Measurement of *R. solanacearum*

Prepare liquid media by adding the extract of *A. annua* to CPG liquid medium to reach final concentrations of 0, 1.5, and 3 g·mL^−1^. Pick a single colony of *R. solanacearum* and inoculate it into the liquid medium containing the extract, and mix thoroughly. Incubate the inoculated medium for 24 h. Then, centrifuge the bacterial suspension at 4000× *g* for 1 min to harvest the bacterial cells. For processing bacterial cells for scanning electron microscope observation, refer to the method proposed by Zhang et al. [59].

### 4.4. Assessment of Disease Resistance in Pepper Plants After Treatment with A. annua Extract

#### 4.4.1. Potted Plant Experiment and Sampling Method

*R. solanacearum* was cultured in liquid CPG medium for 24 h. Subsequently, the bacterial cells were collected by centrifugation at 3000–5000 rpm for 10 min. These cells were then resuspended in sterile water to prepare a bacterial suspension with an OD600 value of 0.1, which served as the inoculum for subsequent experiments. The study comprised five treatments: 0.75 g·mL^−1^, 1.5 g·mL^−1^, and 3 g·mL^−1^ concentrations of *A. annua* extract, a tap water blank control, and a sole *R. solanacearum* inoculation treatment. The procedures for spray application of the extract and for inoculating the pathogen adhered to the methodologies outlined by Zhang et al. [59]. Pepper plants of uniform size and growth vigor were selected, with nine pepper seedlings chosen for each treatment. Firstly, using a scalpel, make three evenly spaced small trenches, each measuring 2 cm in length and 5 cm in depth, on one side of the soil located 2 cm away from the pepper stem. Subsequently, pour the extract and bacterial suspension sequentially into these trenches. In the primary treatment group, 5 mL of the extract was initially administered to the roots, with a subsequent application of 5 mL of *R. solanacearum* suspension three hours later. An additional 5 mL of the extract was administered five days post-inoculation. Conversely, in the blank control group, 5 mL of tap water was initially applied to the roots, followed by another 5 mL of tap water three hours later, and a final 5 mL addition of tap water five days later. In the treatment involving sole inoculation, 5 mL of tap water was first applied to the roots. Three hours later, 5 mL of *R. solanacearum* suspension was added. Subsequently, another 5 mL of tap water was administered five days post-inoculation. Upon completion of the treatment, daily watering with an equivalent amount of tap water was conducted to maintain adequate moisture levels. Daily observations were performed, and once diseased plants were identified, the number of affected plants and the disease severity index were documented. On the 7th and 14th days post-disease onset, a selection of four pepper plants was made from each treatment group. Samples were collected from the upper, middle, and lower leaves of these plants for the quantification of primary metabolites (consisting of soluble sugars, soluble proteins, and free amino acids), defense-related substances (including malondialdehyde (MDA), hydrogen peroxide (H_2_O_2_), flavonoids, and total phenols), as well as the assessment of the activities of defense-associated enzymes (such as superoxide dismutase (SOD), peroxidase (POD), catalase (CAT), polyphenol oxidase (PPO), and phenylalanine ammonia-lyase (PAL).

#### 4.4.2. Measurement of Control Efficacy Against Bacterial Wilt

The number of diseased plants and the disease severity index, which was categorized into levels ranging from 0 to 4 based on the methodology outlined in the literature [62], were recorded. Grade 0: Absence of disease symptoms on the pepper plant. Grade 1: Manifestation of partial wilting in one to two leaves, or chlorotic streaks on the stem covering less than one-third of the plant’s height. Grade 2: Wilting observed in two to three leaves, or chlorotic streaks on the stem spanning between one-third and one-half of the plant’s height. Grade 3: Presence of one to two healthy leaves, or chlorotic streaks on the stem extending from one-half to two-thirds of the plant’s height. Grade 4: Complete absence of healthy leaves on the pepper plant, indicating near-death, or chlorotic streaks on the stem exceeding two-thirds of the plant’s height.

The calculation formula is as follows:Incidence rate = (number of diseased plants/total number of surveyed plants) × 100%Disease index = Σ[(number of diseased plants × value of number of diseased plants)/(total number of surveyed plants × maximum representative value)] × 100

#### 4.4.3. Measurement of Primary Metabolite Content

Measurement of soluble sugar content, soluble protein content, and free amino acid content involves taking 0.1 g of the sample and adding 1 mL of pure water, followed by homogenization in an ice bath. The procedure involves taking 0.1 g of the sample, adding 1 mL of pure water, and homogenizing the mixture in an ice bath. Subsequently, centrifuge the mixture at 8000× *g* for 10 min, and collect the resultant supernatant as the sample solution for further testing. The soluble protein content was determined using the anthrone-sulfuric acid colorimetric method [63], while the soluble protein concentration was assayed via the Coomassie Brilliant Blue method [64]. Place 0.05 g of the sample into a 2 mL centrifuge tube, subsequently add 1 mL of 10% acetic acid, and grind to produce a homogenate. Centrifuge the mixture at 12,000 rpm for 1 min, and thereafter collect the supernatant as the sample solution intended for testing. The ninhydrin colorimetric method, as described by Friedman [65], was employed to determine the content of free amino acids.

#### 4.4.4. Measurement of Defense Substance Content

(1)Measurement of MDA Content

The MDA content was measured using the corresponding kit purchased from Shanghai Yan Zun Biotechnology Co., Ltd., Shanghai, China. To proceed, 0.1 g of the sample is weighed and mixed with 1 mL of extraction buffer. Homogenization is then carried out in an ice bath. Following this, centrifugation is performed at 8000× *g* for 10 min at a temperature of 4 °C. The supernatant is subsequently collected, and a microplate reader sourced from Molecular Devices, San Jose, CA, USA, is utilized to directly measure the absorbance of the pertinent enzyme. Based on this measurement, the enzyme content is calculated.

(2)Measurement of H_2_O_2_ Content

The hydrogen peroxide content was measured using the corresponding kit purchased from Guangzhou Pobo Instrument Co., Ltd., Guangzhou, China. Initially, 0.1 g of the sample is accurately weighed and combined with 1 mL of extraction solution. The resultant mixture is homogenized within an ice bath. Subsequently, centrifugation is conducted at 8000× *g* for a duration of 10 min at a controlled temperature of 4 °C. Following centrifugation, the supernatant is carefully collected and employed for direct absorbance assessment of the pertinent enzyme utilizing a microplate reader. Finally, based on the obtained absorbance values, the enzyme content is determined.

(3)Measurement of total phenol content

To begin, place 0.015 g of the sample into a 2 mL centrifuge tube. Subsequently, add glass beads and a small quantity of quartz sand. Pipette 1 mL of 95% ethanol into the tube and grind the mixture into a homogenate. Finally, determine the total phenol content using the methodology outlined by Feng et al. [66].

(4)Measurement of flavonoid content

Using the sodium nitrite-aluminum nitrate method Following the method of Liu et al. [67], with slight modifications applied. Take 0.015 g of the dried sample and add an appropriate amount of steel balls and a small quantity of quartz sand. Subsequently, pipette in 1 mL of 95% ethanol. Thoroughly grind the mixture until a homogeneous slurry state is achieved. Transfer 0.4 mL of the sample solution into a 10 mL centrifuge tube and adjust the volume to 2.4 mL with water. Subsequently, add 0.4 mL of 5% sodium nitrite, mix thoroughly, and allow the mixture to stand for 6 min. Following this, pipette in 0.4 mL of 10% Al(NO_3_)_3_, mix well again, and let it stand for another 6 min. Then, add 4 mL of 4% Al(NO_3_)_3_ and bring the total volume to 10 mL with water. Finally, shake the mixture thoroughly and allow it to stand for 15 min. Employing 95% ethanol as the reagent blank, determine the absorbance value of the sample at a wavelength of 500 nm, and subsequently utilize this value in the standard curve for computation.

#### 4.4.5. Measurement of Defense-Related Enzyme Activities

The activities of defense enzymes, namely SOD, POD, CAT, PPO, and PAL, were measured using corresponding kits purchased from Shanghai Yanzun Biotechnology Co., Ltd. Add 1 mL of extraction buffer to 0.1 g of pepper leaf samples and homogenize at 4 °C. Centrifuge the mixture at 12,000 rpm for 10 min at 4 °C and collect the supernatant as the test solution. Directly measure the absorbance of the corresponding enzymes using a microplate reader, and calculate their activities.

### 4.5. Data Analysis

All data in this study were analyzed using SPSS 21.0 software for ANOVA and Duncan’s multiple range test (*p* < 0.05). EC_50_ values are calculated using GraphPad Prism 8, while plotting is carried out with Origin 2021.

## 5. Conclusions

*A. annua* exhibits potential applications for the environmentally friendly control of bacterial wilt in peppers. As the concentration of *A. annua* extract increases, the biofilm formation capability of *R. solanacearum* gradually diminishes, accompanied by a progressively severe degree of cell shrinkage. Furthermore, the application of *A. annua* extract significantly decreases both the disease index and incidence rate of bacterial wilt in peppers. On both the 7th and 14th days, applications of *A. annua* extract at concentrations of 0.75 g·mL^−1^ and 3 g·mL^−1^ were observed to elevate the activities of SOD, POD, and CAT in peppers during various growth stages. These treatments also led to a reduction of MDA and H₂O₂ levels, effectively scavenging reactive oxygen species within the plants. Additionally, the extract enhanced the activities of PPO and PAL, and increased the contents of soluble proteins, flavonoids, and total phenols in peppers, ultimately strengthening their disease resistance. Considering the nearly equivalent effects observed for both concentrations, the 0.75 g·mL^−1^ concentration of *A. annua* extract is a more cost-effective choice.

## Figures and Tables

**Figure 1 plants-14-00651-f001:**
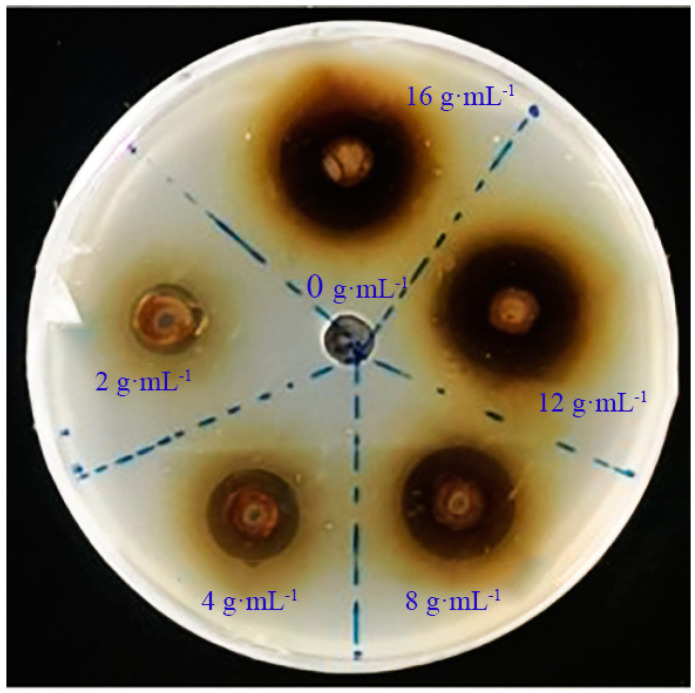
Inhibition zone of *A. annua* extract.

**Figure 2 plants-14-00651-f002:**
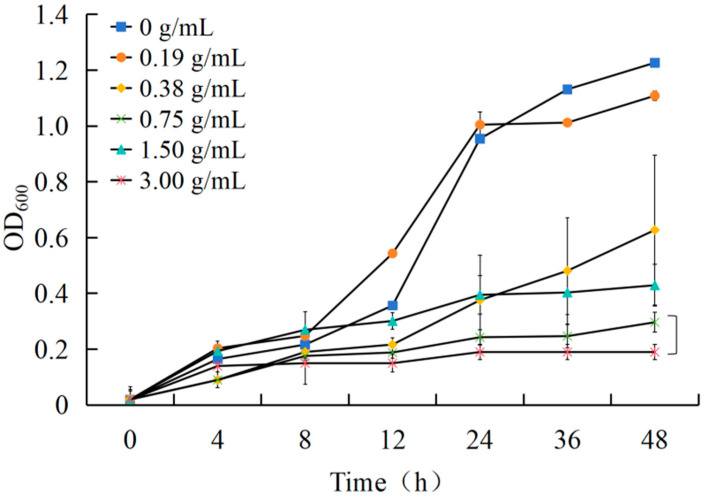
Growth curve of *R. solanacearum* after treatment with *A. annua* extract.

**Figure 3 plants-14-00651-f003:**
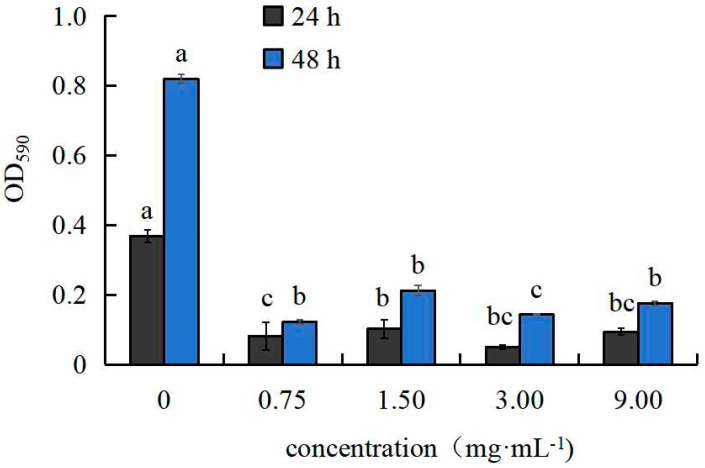
Impact of *A. annua* extract on the formation of *R. solanacearum* biofilm. Different lowercase letters indicate significant differences at the 0.05 level among treatments; the same applies below. The 24-h degrees of F_0.05_(4, 15) = 151.142; the 48-h degrees of F_0.05_(4, 15) = 657.600.

**Figure 4 plants-14-00651-f004:**
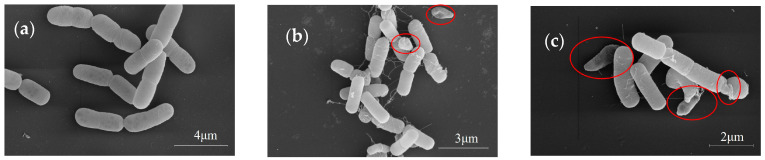
Impact of *A. annua* extract on the morphology of *R. solanacearum* (SEM images). (**a**) 0 g·mL^−1^; (**b**) 1.5 g·mL^−1^; (**c**) 3 g·mL^−1^. The red circles mark the *R. solanacearum* cells that exhibited morphological changes after treatment with *A. annua* extracts.

**Figure 5 plants-14-00651-f005:**
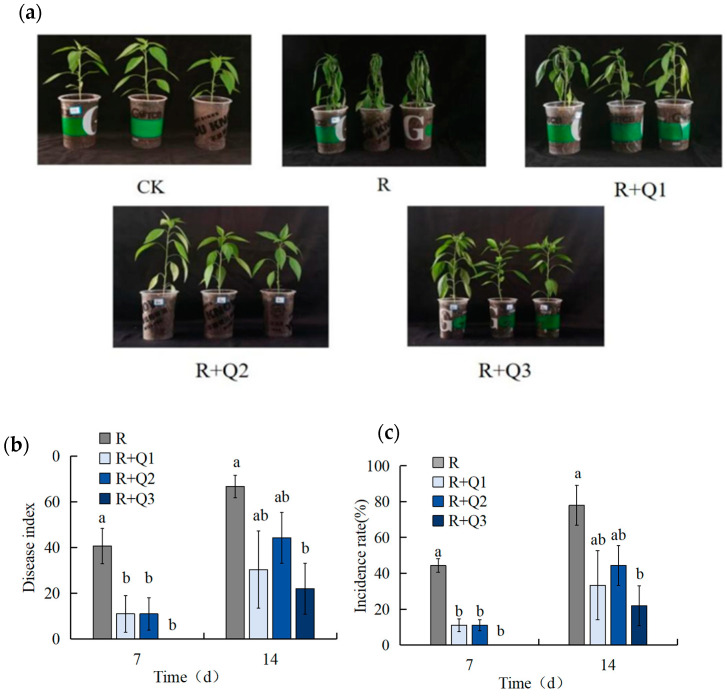
The control effect of *A. annua* extract on pepper wilt disease. (**a**) The control effect of *A. annua* extract on pepper bacterial wilt; (**b**) the disease index of pepper; (**c**) the incidence rate of pepper. Different lowercase letters indicate significant differences at the 0.05 level among treatments. Concentration of bacterial fluid in R OD600 = 0.1; R+Q1 represents 0.75 g·mL^−1^; R+Q2 represents 1.5 g·mL^−1^; R+Q3 represents 3 g·mL^−1^. On the 7th day, the disease index degrees of F_0.05_(3, 8) = 4.052; the incidence rate degrees of F_0.05_(3, 8) = 3.986. On the 14th day, degrees of F_0.05_(3, 8) = 2.718; the incidence rate degrees of F_0.05_(3, 8) = 3.14.

**Figure 6 plants-14-00651-f006:**
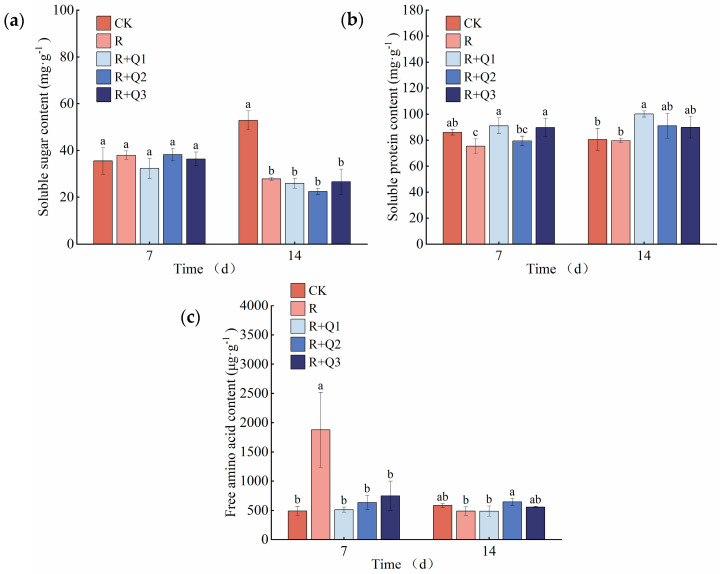
The effect of *A. annua* extract on (**a**) soluble sugar content, (**b**) soluble protein content, and (**c**) free amino acid content in pepper leaves. Different lowercase letters indicate significant differences at the 0.05 level among treatments. CK is a blank treatment; the concentration of bacterial fluid in R OD600 = 0.1; R+Q1 represents 0.75 g·mL^−1^; R+Q2 represents 1.5 g·mL^−1^; R+Q3 represents 3 g·mL^−1^, the same below. On the 7th day, the F values for soluble sugars, soluble proteins, and free amino acids were F_0.05_(4, 10) = 1.198, F_0.05_(4, 10) = 4.999, and F_0.05_(4, 10) = 10.251. On the 14th day, the F values for soluble sugars, soluble proteins, and free amino acids were F_0.05_(4, 10) = 26.778, F_0.05_(4, 10) = 4.432, and F_0.05_(4, 10) = 3.864.

**Figure 7 plants-14-00651-f007:**
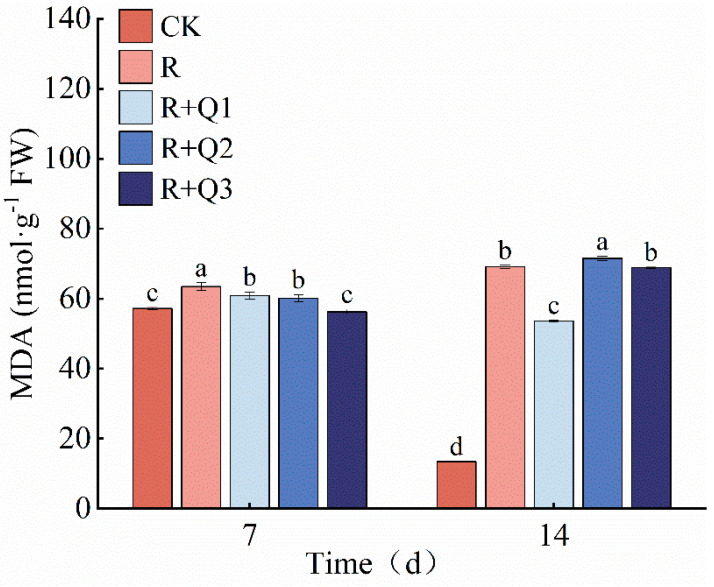
The effect of *A. annua* extract on MDA content in peppers. Different lowercase letters indicate significant differences at the 0.05 level among treatments. On the 7th day, the F value for MDA was F_0.05_(4, 10) = 31.822. On the 14th day, the F value for MDA was F_0.05_(4, 10) = 12750.831.

**Figure 8 plants-14-00651-f008:**
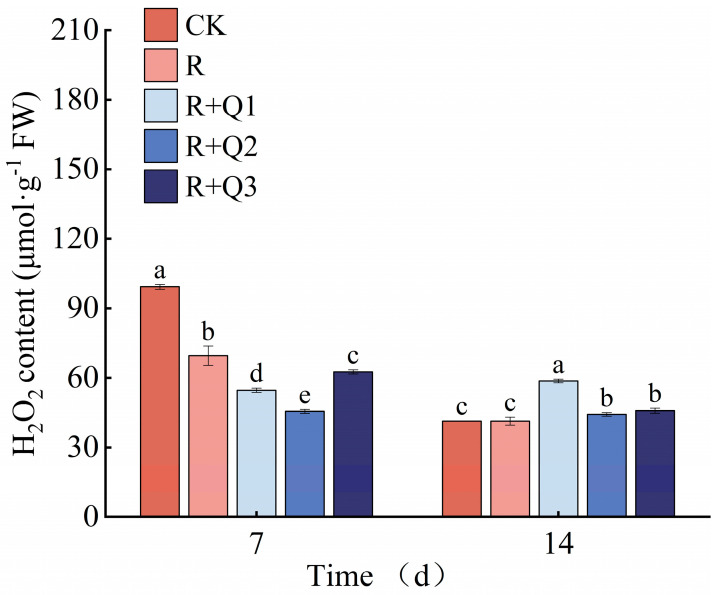
The effect of *A. annua* extract on the H_2_O_2_ content in pepper. Different lowercase letters indicate significant differences at the 0.05 level among treatments. On the 7th day, the F value for H_2_O_2_ was F_0.05_(4, 10) = 301.501. On the 14th day, the F value for H_2_O_2_ was F_0.05_(4, 10) = 144.217.

**Figure 9 plants-14-00651-f009:**
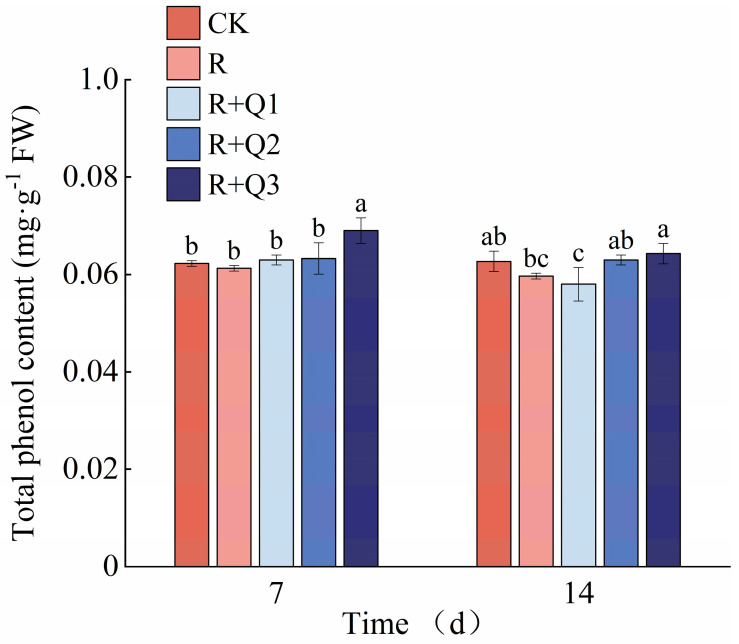
The effect of *A. annua* extract on the total phenol content in pepper. Different lowercase letters indicate significant differences at the 0.05 level among treatments. On the 7th day, the F value for the total phenol was F_0.05_(4, 10) = 7.132. On the 14th day, the F value for the total phenol was F_0.05_(4, 10) = 4.644.

**Figure 10 plants-14-00651-f010:**
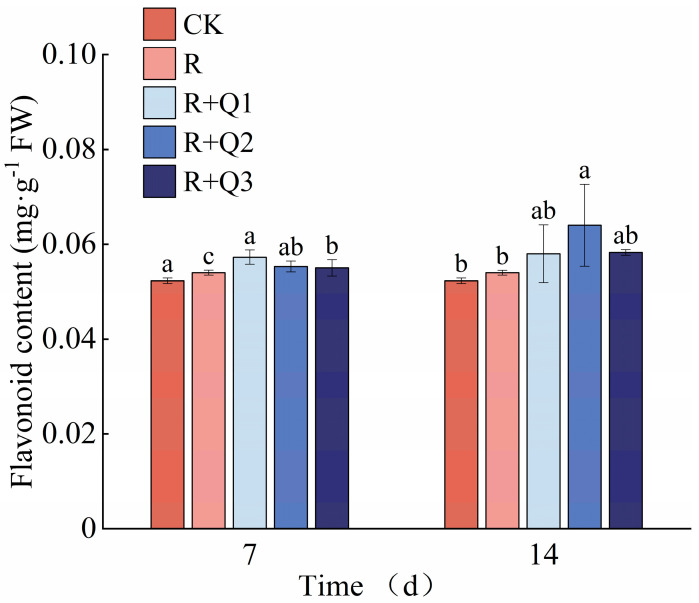
The Effect of *A. annua* extract on the flavonoid content in pepper. Different lowercase letters indicate significant differences at the 0.05 level among treatments. On the 7th day, the F value for the flavonoid was F_0.05_(4, 10) = 7.214. On the 14th day, the F value for the flavonoid was F_0.05_(4, 10) = 2.729.

**Figure 11 plants-14-00651-f011:**
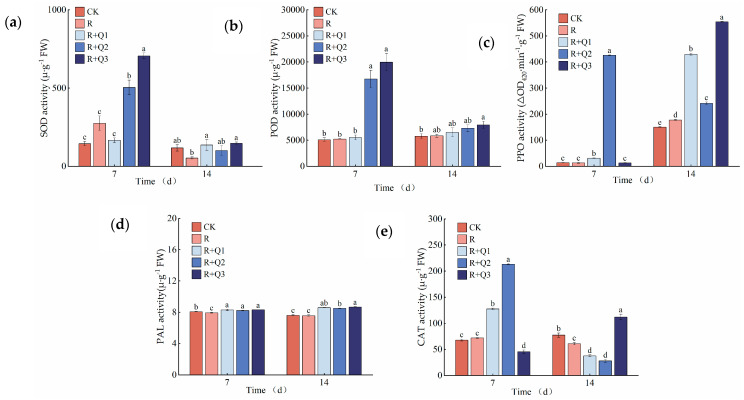
Effects of *A. annua* extract on the activity of defense-related enzymes in pepper leaves. (**a**) The effect of *A. annua* extract on the activity of SOD in pepper leaves; (**b**) the effect of *A. annua* extract on the activity of POD in pepper leaves; (**c**) the effect of *A. annua* extract on the activity of CAT in pepper leaves; (**d**) the effect of *A. annua* extract on the activity of PAL in pepper leaves; (**e**) the effect of *A. annua* extract on the activity of PPO in pepper leaves. Different lowercase letters indicate significant differences at the 0.05 level among treatments. On the 7th day, the F values for SOD, POD, PPO, PAL, and CAT were F_0.05_(4, 10) = 0.456, F_0.05_(4, 10) = 44.050, F_0.05_(4, 10) = 67783.874, F_0.05_(4, 10) = 25.314, and F_0.05_(4, 10) = 1293.564. On the 14th day, the F values for SOD, POD, PPO, PAL, and CAT were F_0.05_(4, 10) = 2.614, F_0.05_(4, 10) = 2.237, F_0.05_(4, 10) = 9745.103, F_0.05_(4, 10) = 269.154, and F_0.05_(4, 10) = 92.257.

**Table 1 plants-14-00651-t001:** Diameter of inhibition zone of *A. annua* extract, EC_50_ computed by GraphPad Prism 8. Different lowercase letters indicate the significant differences between different samples (*p* < 0.05). The same below. F_0.05_(5, 15) = 2116.927.

Concentration (g·mL^−1^)	Diameter of Inhibition Zone (mm)	Inhibition Zone	Virulence Regression Equation	EC_50_ Value
0	0	−		
2	4.07 ± 0.09 e	+	y = 0.7551x + 4.6811 R^2^ = 0.9856	2.66 g·mL^−1^
4	6.71 ± 0.05 d	+		
8	9.21 ± 0.16 c	+		
12	6.71 ± 0.05 d	++		
16	12.62 ± 0.16 a	++		

The symbol “+”, “++” and “−” in the table represent the diameters of the inhibition zones.

**Table 2 plants-14-00651-t002:** MIC values of *A. annua* extract.

Concentration (g·mL^−1^)	Phenomenon
0.19	+
0.38	+
0.75	+
1.5	+
3	−
6	−

The symbol “−” in the table signifies the absence of red coloration and no growth of *R. solanacearum*, whereas “+” indicates the presence of red coloration, indicative of *R. solanacearum*.

## Data Availability

Data are contained within the article.

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
