# Peer review of "Bacteriostatic Activity and Resistance Mechanism of Artemisia annua Extract Against Ralstonia solanacearum in Pepper"

_plants, 2025, doi:10.3390/plants14050651_

Round 1
Reviewer 1 Report
Comments and Suggestions for Authors
Only suggestion is that clarity of the Figures should be improved as well as visibility of colors and units in them.
Author Response
Point 1: Only suggestion is that clarity of the Figures should be improved as well as visibility of colors and units in them.
Response 1: Thank you for your suggestions. We have modified the clarity and size of the numbers in the figures of the article, such as enhancing the clarity of the numbers in Figure 1 and improving the clarity of the scale bar in Figure 4. We sincerely thank you for your careful reading. Wishing you good health and a happy life.

Reviewer 2 Report
Comments and Suggestions for Authors
The manuscript gives new information about bacteriostatic activity of Artemisia annua extract in pepper. But I think the manuscript has some problems and it requires “major revisions” for publication.
1. L.62-64 (L22-24)
Authors described advantages of plant-derived pesticides hear. They should add citations.
2. L.29, L.31
7th day, 14th day → 7th day after ○, 14th day after ○
3. 2.1 The inhibitory effect of Artemisia annua on R. solanacearum activity (L.94-99) & 4.4.1 Solid plate assay of Artemisia annua extract for inhibiting R. solanacearum (L.399-)
Authors describe that “40 uL of sterile water” was used in this test (L408-409). They should add the results of this treatment.
Authors should add the calculation method of “EC50”.
4. 2.2. MIC value & 4.2.2 The determination of minimum inhibitory concentration (L.418-)
Authors should match the “concentration” of Artemisia annua extract between table 2 and methods.
5. L101
MIC → minimum inhibitory concentration (MIC)
6. Effect of Artemisia annua extract
Authors describe that 3 g/mL-1 showed inhibitory of R. solanacearum (L107). But in inhibition curve test and effect on the biofilm test, less concentration of extract showed inhibitory effect. Authors should discuss most effective concentration comprehensively.
7. biofilm test
Authors describe that incubate the plates ’for 72 h’ in the method section (L.446). They should add the result of this treatment.
8. Figure 3
Authors should change polygonal line graph to other graph.
9. effect of morphology
Authors should describe where “shrinkage” or “denaturation” or “depression” are shown using Figure 4.
10. Figure 4
Scale bar is unclear.
There is no description about “CK”.
11. efficacy in controlling pepper wilt disease, impact of extract on contents
Authors should describe of effect of Artemisia annua extract using value of disease index, incidence rate or contents but not reduction rate.
12. L. 152
“sugar” → “ protein”
Author Response
Point 1: L.62-64 (L22-24)
Authors described advantages of plant-derived pesticides hear. They should add citations.
Response 1: Thank you for your suggestions. We have added citation [12].
Point 2: L.29, L.31
7th day, 14th day → 7th day after ○, 14th day after ○
Response 2: Thank you for your suggestions. We have made modifications within the text.
Point 3: 2.1 The inhibitory effect of Artemisia annua on R. solanacearum activity (L.94-99) & 4.4.1 Solid plate assay of Artemisia annua extract for inhibiting R. solanacearum (L.399-)
Authors describe that “40 uL of sterile water” was used in this test (L408-409). They should add the results of this treatment.
Authors should add the calculation method of “EC50”.
Response 3: Thank you for your suggestions. We have added the experimental results for CK in Table 1, and also noted that the EC50 values were calculated using GraphPad Prism 8.
Point 4: 2.2. MIC value & 4.2.2 The determination of minimum inhibitory concentration (L.418-)
Authors should match the “concentration” of Artemisia annua extract between table 2 and methods.
Response 4: Thank you for your suggestions. We have made modifications within the text.
Point 5: L101
MIC → minimum inhibitory concentration (MIC)
Response 5: Thank you for your suggestions. We have made modifications within the text.
Point 6: Effect of Artemisia annua extract
Authors describe that 3 g/mL-1 showed inhibitory of R. solanacearum (L107). But in inhibition curve test and effect on the biofilm test, less concentration of extract showed inhibitory effect. Authors should discuss most effective concentration comprehensively.
Response 6: Thank you for your suggestions. The reasons for selecting 3 as the MIC value are as follows: (1) According to the 2.2 MIC test results, it can be concluded that a complete inhibition of Ralstonia solanacearum growth can be achieved at a concentration of 3 g·mL⁻¹ of the artemisia extract. (2) The results of the antibacterial curve experiment showed that, at a concentration of 3 g·mL⁻¹ of the artemisia extract, the growth of Ralstonia solanacearum was completely inhibited after 24 hours. (3) Additionally, during the biofilm assay, at a concentration of 3 g·mL⁻¹ of the artemisia extract, the ability of Ralstonia solanacearum to form biofilms was inhibited. Therefore, in summary, it can be seen that the artemisia extract at a concentration of 3 g·mL⁻¹ is more suitable as the MIC value. However, it is necessary to optimize and improve the experimental environment and operational procedures described in the manuscript to ensure the accuracy of the data.
Point 7: biofilm test
Authors describe that incubate the plates ’for 72 h’ in the method section (L.446). They should add the result of this treatment.
Response 7: Thank you for your suggestions. The biofilm assay was only conducted for 24 hours and 48 hours, and we have removed the "72 hours" mention from the methods section.
Point 8: Figure 3
Authors should change polygonal line graph to other graph.
Response 8: Thank you for your suggestions. We have changed the line graph to a bar chart.
Point 9: effect of morphology
Authors should describe where “shrinkage” or “denaturation” or “depression” are shown using Figure 4.
Response 9: Thank you for your suggestions. We have highlighted the areas where bacterial deformation occurs in the figure with prominent red circles, and have also provided a more detailed description in the results section.
Point 10: Figure 4
Scale bar is unclear.
There is no description about “CK”.
Response 10: Thank you for your suggestions. We have improved the clarity of the scale bar and have re-described the results accordingly. “The morphological alterations of R. solanacearum were observed following treatment with A. annua extract at varying concentrations. In the control group (CK), the bacterial cells maintained their normal, turgid morphology without any observable deformations. However, treatment with A. annua extracts induced notable morphological alterations, including cell shrinkage, deformation, and surface depression. These morphological changes exhibited a concentration-dependent response, with more pronounced shrinkage, deformation, and depression observed at higher extract concentrations (Figure 4).”
Point 11: efficacy in controlling pepper wilt disease, impact of extract on contents
Authors should describe of effect of Artemisia annua extract using value of disease index, incidence rate or contents but not reduction rate.
Response 11: Thank you for your suggestions. We have made modifications within the text. “The extract from A. annua could significantly reduce the disease index and incidence of pepper bacterial wilt, which was attributed to R. solanacearum. On the 7th day, the disease index and incidence rate in the treatment solely involving R. solanacearum were 44.33 and 40.67%, respectively. In contrast, the treatments with 0.75 g·m⁻¹, 1.50 g·m⁻¹, and 3.00 g·m⁻¹ extracts of A. annua exhibited disease indices and incidence rates of 14.29 and 10%, 14.29 and 10%, and 0%, respectively. On the 14 day, the disease index and incidence rate for the treatment solely involving R. solanacearum were 66.67% and 78%, respectively. In comparison, the treatments with 0.75 g·mL⁻¹, 1.50 g·mL⁻¹, and 3.00 g·mL⁻¹ extracts of A. annua exhibited disease indices of 25%, 42.85%, and 21.42%, respectively, and incidence rates of 30.33%, 44.33%, and 22%. These findings demonstrate that the application of A. annua extracts resulted in a notable decrease in both the disease index and incidence rate compared to the R. solanacearum-only treatment. (Figure 5).”
Point 12: L. 152
“sugar” → “ protein”
Response 12: Thank you for your suggestions. We have made modifications within the text.

Reviewer 3 Report
Comments and Suggestions for Authors
Dear authors, your study on the use of Artemisia annua extracts to control the growth of Ralstonia solanacearum is highly interesting and well-described. The methodology is clearly outlined, and the results are presented in a compelling manner, providing valuable insights into the antimicrobial potential of plant-derived compounds. A few minor revisions are included in the document sent as attachment
Best regards,

Author Response
Point 1: L.3: Ralstonia solanacearum
Response 1: Thank you for your suggestions. We have changed “Ralstonia solanacearum” to “ Ralstonia solanacearum”.
Point 2: please change in plant-derived.
Response 2: Thank you for your suggestions. We have changed “Compared with traditional chemical 22 pesticides, Plant-derived pesticides have the advantages of high efficiency, no residue, 23 low toxicity, broad spectrum and low resistance.” to “Plant-derived pesticides exhibit advantages of high efficiency and broad spectrum when compared to traditional chemical pesticides.”
Point 3: L.24: took Artemisia annua as the 24 material to
Response 3: Thank you for your suggestions. We have made modifications within the text.
Point 4: L.25: explored
Response 4: Thank you for your suggestions. We have made modifications within the text.
Point 5: L.27: small letter…extracts.
Response 5: Thank you for your suggestions. We have changed “The results showed that the Minimum Inhibitory Concentration (MIC) of the Artemisia annua extract was 3 g·mL⁻¹; Extracts of high concentration can inhibit the formation of biofilm in Ralstonia solanacearum, leading to the shrinkage of bacterial cells.” to “The results showed that the Minimum Inhibitory Concentration (MIC) of the A. annua extract was 3 g·mL⁻¹; As the concentration of A. annua extract increased, the extent of R. solanacearum cell crumpling intensified, accompanied by a gradual decline in its bio-film-forming ability.”
Point 6: L.40: greater
Response 6: Thank you for your suggestions. We have changed “the application of A. annua extract at a concentration of 0.75g·mL⁻¹ demonstrates greater potential for green control measures against bacterial wilt in peppers.” to “the application of A. annua extract at a concentration of 0.75g·mL⁻¹ demonstrates great potential for green control measures against bacterial wilt in peppers.”
Point 7: L.51: please, when at the beginning of a sentence, latin names should be written as full name= Ralstonia.
Response 7: Thank you for your suggestions. We have changed “R. solanacearum” to “Ralstonia solanacearum”.
Point 8: L.71: whose dried aerial parts are harvested,
Response 8: Thank you for your suggestions. We have changed “Artemisia annua L, whose dried aerial parts are harvested, belongs to the Asteraceae family. Among its components, artemisinin has significant antimalarial effects and is an important ingredient in global antimalarial drugs.” to “The dried aerial parts of the composite plant Artemisia annua L. constitute A. annua, whose primary constituent, artemisinin, demonstrates significant antimalarial activity and functions as a vital component in global antimalarial medications.”
Point 9: L.75: Artemisia
Response 9: Thank you for your suggestions. We have changed “Artemisia annua” to “A. annua”.
Point 10: L.75: .
Response 10: Thank you for your suggestions. We have changed “Artemisia annua” to “A. annua”.
Point 11: L.76: Similarly, the
Response 11: Thank you for your suggestions. We have changed “The ethanol extract of A. annua has bacteriostatic activity against the pathogen of tomato late blight, and its MIC is 15.63 mg·mL⁻¹” to “Similarly the ethanol extract of A. annua has bacteriostatic activity against the pathogen of tomato late blight, and its MIC is 15.63 mg·mL⁻¹.”
Point 12: L.79: Artemisia
Response 12: Thank you for your suggestions. We have changed “Artemisia annua” to “A. annua”.
Point 13: L.79: .
Response 13: Thank you for your suggestions. We have changed “Artemisia annua” to “A. annua”.
Point 14: L.80: Artemisia
Response 14: Thank you for your suggestions. We have changed “Artemisia annua” to “A. annua”.
Point 15: L.80:.
Response 15: Thank you for your suggestions. We have changed “Artemisia annua” to “A. annua”.
Point 16: L.87: as above A. annua.
Response 16: Thank you for your suggestions. We have changed “Artemisia annua” to “A. annua”.
Point 17: L.89: as above
Response 17: Thank you for your suggestions. We have changed “Artemisia annua” to “A. annua”.
Point 18: L.91: as above
Response 18: Thank you for your suggestions. We have changed “Artemisia annua” to “A. annua”.
Point 19: L.97: Zone to zone.
Response 19: Thank you for your suggestions. We have made modifications within the text.
Point 20: L.99: zone
Response 20: Thank you for your suggestions. We have made modifications within the text.
Point 21: L.111: the quality of the figure is low.
Response 21: Thank you for your suggestions. We have made modifications within the text.
Point 22: L.119: please specify both in table and figures the meanning of the small letters e.g
Different letters indicate a significance difference among levels of concentrations (P, 0.05).
Response 22: Thank you for your suggestions. We have made modifications within the text.
Point 23: L.124: please specify they are SEM pics.
Response 23: Thank you for your suggestions. We have changed “Figure 4. Impact of A. annua extract on the morphology of R. solanacearum. (a) 0 g·mL-1; (b) 1.5 g·mL-1; (c) 3 g·mL-1.” to “Figure 4. Impact of A. annua extract on the morphology of R. solanacearum (SEM pics). (a) 0 g·mL-1; (b) 1.5 g·mL-1; (c) 3 g·mL-1.”
Point 24: L.141: please wirte Time (d) as Figure 6 or viceversa.
Response 24: Thank you for your suggestions. We have changed "time/d" in Figure 5 to "time(d)".
Point 25-26: L.144: “Concentration of bacterial fluid in R OD600”. this sentence could be deleted.
Response 25: Thank you for your suggestions. We have changed “The control effect of Artemisia annua extract on pepper wilt disease. (a) the control effect of Artemisia annua extract on pepper bacterial wilt; (b) the disease index of pepper; (c) the incidence rate of pepper. Concentration of bacterial fluid in R OD600 = 0.1;R+Q1 represents 0.75 g·mL-1; R+Q2 represents 1.5 g·mL-1; R+Q3 represents 3 g·mL-1.” To “The control effect of A. annua extract on pepper wilt disease. (a) the control effect of A. annua extract on pepper bacterial wilt; (b) the disease index of pepper; (c) the incidence rate of pepper. Conce-tration of bacterial fluid in R OD600= 0.1;R+Q1 represents 0.75 g·mL-1; R+Q2 represents 1.5 g·mL-1; R+Q3 represents 3 g·mL-1. On the 7th day, the disease index degrees of F0.05(3,8)= 4.052, the inci-dence rates degrees of F0.05(3,8)= 3.986. On the 14th day degrees of F0.05(3,8)= 2.718, the incidence rates degrees of F0.05(3,8)= 3.14.”
Point 27: L.344: strain of.
Response 26: Thank you for your suggestions. We have changed “The R. solanacearum was provided by the Population Microbiology Research Center of the Microbiology Laboratory at South China Agricultural University, and was isolated from R. solanacearum EP1 in Guangdong Province.” to “The strain of R. solanacearum was provided by the Population Microbiology Research Center of the Microbiology Laboratory at South China Agricultural University, and was isolated from R. solanacearum EP1 in Guangdong Province.”
Point 28: L.349: The
Response 27: Thank you for your suggestions. We have deleted it.

Reviewer 4 Report
Comments and Suggestions for Authors
These are my main comments on the manuscript (planst-3468849) entitled “Bacteriostatic activity and resistance mechanism of Artemisia annua extract against Ralstonia solanacearum in pepper”. This work investigates the antibacterial activity of the extract of Artemisia annua against Ralstonia solanacearum in pepper. However, details about introduction, materials and methods, results, and conclusion section are needed. Following substantial revisions should be incorporated in the manuscript prior to acceptance.
A few points:
Ls.22-24: Revise this sentence to eliminate rewordiness.
L.26: A brief sentence about methodology is needed in abstract.
Ls.27, 28, 30, 32, 39-40, etc.: Write out scientific names in full for the first time. Then use the abbreviation A. annua or R. solanacearum. Correct in all manuscript.
L.28: …the biofilm formation in…
L.23: Keywords should be in alphabetic order. Also, keywords serve to widen the opportunity to be retrieved from a database. To put words that already are into title and abstracts makes KW not useful. Please choose terms that are neither in the title nor in abstract.
Ls.46-48: Revise this sentence to eliminate rewordiness.
Ls.48-49: Again, revise this sentence to eliminate rewordiness.
L.57: Delete “mainly”.
L.60: Delete “highly”.
Ls.62-64: Revise this sentence to eliminate rewordiness.
Ls.71-73: Again, revise this sentence to eliminate rewordiness.
Ls.86-88: Information about Ralstonia solanacearum in pepper is needed.
L.90: Also, a hypothesis for this study is needed.
Ls.94-235: Statistical results are missing. For each ANAVA, provide the F-value, degree freedom and p-value.
Ls.100-104: For data analysis, any statistical method?
Ls.120-123: Provide more details of this result.
L.124: CK is the control?
Ls.237-244: This information should be in introduction section.
Ls.246-248: This information should be in conclusions section.
Ls.265-267: Summarize this sentence.
Ls.267-269: Revise this sentence to eliminate rewordiness.
L.274: Botritys should be in italic.
Ls.277-278: Revise this sentence to eliminate rewordiness.
L.339: R. solanacearum should be in italic.
L.365: Roomtemperature? Revise.
Ls.395-398: This information should be in results section.
L.405 and 468: Delete “approximately”.
Ls.419-422: Rephrase this sentence.
Ls.563-564: Duncan's multiple range test is not rigorous; I suggest replace by Tukey HSD test.
Author Response
Point 1: Ls.22-24: Revise this sentence to eliminate rewordiness.
Response 1: Thank you for your suggestions. We have revised the sentence to “Plant-derived pesticides exhibit advantages of high efficiency and broad spectrum when compared to traditional chemical pesticides.”
Point 2: L.26: A brief sentence about methodology is needed in abstract.
Response 2: Thank you for your suggestions. We have revised the sentence to “Artemisia annua and 'Tai Jiao' No. 1 were used as the experimental materials, and treated with 0.75 g·mL-1, 1.5 g·mL-1, 3 g·mL-1 of A. annua extract and inoculated with R. solanacearum at a concentration of OD600=0.1 for 14 days.”
Point 3: Ls.27, 28, 30, 32, 39-40, etc.: Write out scientific names in full for the first time. Then use the abbreviation A. annua or R. solanacearum. Correct in all manuscript.
Response 3: Thank you for your suggestions. We have made modifications within the text.
Point 4: L.28: …the biofilm formation in…
Response 4: Thank you for your suggestions. We have made modifications within the text.
Point 5: L.23: Keywords should be in alphabetic order. Also, keywords serve to widen the opportunity to be retrieved from a database. To put words that already are into title and abstracts makes KW not useful. Please choose terms that are neither in the title nor in abstract.
Response 5: Thank you for your suggestions. We have revised the sentence to “Artemisia annua; botanical pesticide; control effect; Ralstonia solanacearum; resistance mechanis”.
Point 6: Ls.46-48: Revise this sentence to eliminate rewordiness.
Response 6: Thank you for your suggestions. We have revised the sentence to “Pepper serves as a prominent cash crop in China, with a cultivation area exceeding 1,533,300 hm2 in 2021. This figure accounts for 40% of the global cultivation area, thereby establishing it as the most extensively grown vegetable in China.”
Point 7: Ls.48-49: Again, revise this sentence to eliminate rewordiness.
Response 7: Thank you for your suggestions. We have revised the sentence to “Pepper bacterial wilt, caused by Ralstonia solanacearum, is a soil-borne bacterial disease distinguished by its broad host adaptability, potent latent infectivity, and significant genetic variability, which pose severe impediments to the advancement of the pepper industry.”
Point 8: L.57: Delete “mainly”.
Response 8: Thank you for your suggestions. We have made modifications within the text.
Point 9: L.60: Delete “highly”.
Response 9: Thank you for your suggestions. We have made modifications within the text.
Point 10: Ls.62-64: Revise this sentence to eliminate rewordiness.
Response 10: Thank you for your suggestions. We have revised the sentence to “Plant-derived pesticides exhibit advantages of high efficiency and broad spectrum when compared to traditional chemical pesticides.”
Point 11: Ls.71-73: Again, revise this sentence to eliminate rewordiness.
Response 11: Thank you for your suggestions. We have revised the sentence to “The dried aerial parts of the composite plant Artemisia annua L. constitute A. annua, whose primary constituent, artemisinin, demonstrates significant antimalarial activity and functions as a vital component in global antimalarial medications.”
Point 12: Ls.86-88: Information about Ralstonia solanacearum in pepper is needed.
Response 12: Thank you for your suggestions. We have revised the sentence to “Ralstonia solanacearum possesses the ability to survive for extended periods in soil or plant residues. The combination of its robust infectivity and broad host range poses significant challenges in achieving complete control of bacterial wilt in agricultural settings. Moreover, as of the present date, no plant-derived fungicide has been developed specifically for the management of pepper bacterial wilt.”
Point 13: L.90: Also, a hypothesis for this study is needed.
Response 13: Thank you for your suggestions. We have revised the sentence to “Therefore, the development of plant-based fungicides specifically aimed at preventing and controlling bacterial wilt in peppers can provide new options for the future management of this disease. Given the remarkable antimicrobial capabilities of A. annua, we formulate the hypothesis that it may exhibit inhibitory effects against R. solanacearum as well.”
Point 14: Ls.94-235: Statistical results are missing. For each ANAVA, provide the F-value, degree freedom and p-value.
Response 14: Thank you for your suggestions. We have made modifications within the text. For example “Figure 5. The control effect of A. annua extract on pepper wilt disease. (a) the control effect of A. annua extract on pepper bacterial wilt; (b) the disease index of pepper; (c) the incidence rate of pepper. Concentration of bacterial fluid in R OD600= 0.1;R+Q1 represents 0.75 g·mL-1; R+Q2 represents 1.5 g·mL-1; R+Q3 represents 3 g·mL-1. On the 7th day, the disease index degrees of F0.05(3,8)= 4.052, the incidence rates degrees of F0.05(3,8)= 3.986. On the 14th day degrees of F0.05(3,8)= 2.718, the incidence rates degrees of F0.05(3,8)= 3.14.”
Point 15: Ls.100-104: For data analysis, any statistical method?
Response 15: Thank you for your suggestions. The primary statistical method utilized in the results of this experiment is one-way ANOVA (Analysis of Variance).
Point 16: Ls.120-123: Provide more details of this result.
Response 16: Thank you for your suggestions. We have revised the sentence to “The morphological alterations of R. solanacearum were observed following treatment with A. annua extract at varying concentrations. In the control group (CK), the bacterial cells maintained their normal, turgid morphology without any observable deformations. However, treatment with A. annua extracts induced notable morphological alterations, including cell shrinkage, deformation, and surface depression. These morphological changes exhibited a concentration-dependent response, with more pronounced shrinkage, deformation, and depression observed at higher extract concentrations (Figure 4).”
Point 17: L.124: CK is the control?
Response 17: Thank you for your suggestions. We have changed CK to 0 g·mL-1.
Point 18: Ls.237-244: This information should be in introduction section.
Response 18: Thank you for your suggestions. We have made modifications within the text.
Point 19: Ls.246-248: This information should be in conclusions section.
Response 19: Thank you for your suggestions. We have made modifications within the text.
Point 20: Ls.265-267: Summarize this sentence.
Response 20: Thank you for your suggestions. We have revised the sentence to “The natural existence of plants inevitably leads to their encounter with a diverse array of pathogenic microorganisms in their environment. Through the protracted process of interacting and adapting to these pathogens, plants have progressively evolved intricate and highly sophisticated defense systems, aimed at minimizing and mitigating the threats posed by these pathogens.”
Point 21: Ls.267-269: Revise this sentence to eliminate rewordiness.
Response 21: Thank you for your suggestions. We have revised the sentence to “In the context of plant defense systems, a close correlation has been observed between primary and secondary metabolites, wherein the latter are utilized for defensive purposes.”
Point 22: L.274: Botritys should be in italic.
Response 22: Thank you for your suggestions. We have made modifications within the text.
Point 23: Ls.277-278: Revise this sentence to eliminate rewordiness.
Response 1: Thank you for your suggestions. We have made modifications within the text.
Point 23: L.339: R. solanacearum should be in italic.
Response 1: Thank you for your suggestions. We have made modifications within the text.
Point 24: L.365: Roomtemperature? Revise.
Response 24: Thank you for your suggestions. We have made modifications within the text.
Point 25: Ls.395-398: This information should be in results section.
Response 25: Thank you for your suggestions. We have made modifications within the text.
Point 26: L.405 and 468: Delete “approximately”.
Response 26: Thank you for your suggestions. We have made modifications within the text.
Point 27: Ls.419-422: Rephrase this sentence.
A serial double-dilution method was employed to prepare six concentration gradients of the aqueous extract of A. annua in TTC liquid medium, namely 8 g/mL, 4 g/mL, 2 g/mL, 1 g/mL, 0.5 g/mL, and 0.25 g/mL.
Response 27: Thank you for your suggestions. We have revised the sentence to “I A serial double-dilution method was employed to prepare six concentration gradients of the aqueous extract of A. annua in TTC liquid medium, namely 0.19 g·mL⁻¹, 0.38 g·mL⁻¹, 0.75 g·mL⁻¹, 1.5 g·mL⁻¹, 3 g·mL⁻¹, and 6 g·mL⁻¹.”
Point 28: Ls.563-564: Duncan's multiple range test is not rigorous; I suggest replace by Tukey HSD test.
Response 28: Thank you for your suggestions. You have opened up a new perspective for me in data processing. Upon reviewing the website, I found that the Tukey HSD test statistical method is more accurate than Dunn's test in certain situations. However, since I am not familiar with the Tukey HSD test and it will take some time for me to learn how to apply it, I did not make any modifications to the article based on this finding. Nevertheless, I can assure you that our data processing is authentic and reliable. Thank you very much for your suggestion.

Round 2
Reviewer 2 Report
Comments and Suggestions for Authors
Dear authors,
Thank you for revising the manuscript. I think the manuscript should be revised in only few points.
- L.29, L.31
7th day, 14th day → 7th day after “inoculation” or “treatment”
- effect of morphology
Authors should describe meaning of ‘red circles’.
Author Response
Point 1: L.29, L.31
7th day, 14th day → 7th day after “inoculation” or “treatment”
Response 1: Thank you for your suggestions. We have changed “On the 14th day after” and “At both the 7th and 14th days after” to “On the 14th day after treatment” and “At both the 7th and 14th days after treatment”.
Point 2: effect of morphology
Authors should describe meaning of ‘red circles’.
Response 2: Thank you for your suggestions. We have described it.
Reviewer 4 Report
Comments and Suggestions for Authors
The authors have incorporated all suggestions and comments into the revised version, now the manuscript seems much clear. There are some minor points to be corrected:
The authors have incorporated all suggestions and comments into the revised version, now the manuscript seems much clear. There are some minor points to be corrected:
1. Keywords serve to widen the opportunity to be retrieved from a database. To put words that already are into title and abstracts makes KW not useful. Please choose terms that are neither in the title nor in abstract.
2. See Introduction (second paragraph): The dried aerial parts of the composite plant Artemisia annua L. constitute A. annua, whose primary constituent, artemisinin, demonstrates significant antimalarial activity and functions as a vital component in global antimalarial medications...Please, summarized this sentence.
Author Response
Point 1: Keywords serve to widen the opportunity to be retrieved from a database. To put words that already are into title and abstracts makes KW not useful. Please choose terms that are neither in the title nor in abstract.
Response 1: Thank you for your suggestions. We have changed “Artemisia annua; botanical pesticide; control effect; Ralstonia solanacearum; resistance mechanism” to “bacterial wilt; bio-control; antibacterial activity; enzyme activity; plant extracts”.
Point 2: See Introduction (second paragraph): The dried aerial parts of the composite plant Artemisia annua L. constitute A. annua, whose primary constituent, artemisinin, demonstrates significant antimalarial activity and functions as a vital component in global antimalarial medications...Please, summarized this sentence.
Response 2: Thank you for your suggestions. We have changed “The dried aerial parts of the composite plant Artemisia annua L. constitute A. annua, whose primary constituent, artemisinin, demonstrates significant antimalarial activity and functions as a vital component in global antimalarial medications.” to “The dried aerial parts of the plant Artemisia annua contain artemisinin, its main compound, which has strong antimalarial properties and is a key ingredient in global antimalarial drugs.”